# Real-Time Monitoring of Lysosomal Membrane Permeabilization Using Acridine Orange

**DOI:** 10.3390/mps6040072

**Published:** 2023-08-09

**Authors:** Ida Eriksson, Linda Vainikka, Hans Lennart Persson, Karin Öllinger

**Affiliations:** 1Experimental Pathology, Department of Biomedical and Clinical Sciences, Linköping University, 581 85 Linköping, Sweden; linda.vainikka@liu.se (L.V.); karin.ollinger@liu.se (K.Ö.); 2Department of Respiratory Medicine in Linköping, Linköping University, 581 85 Linköping, Sweden; lennart.persson@liu.se; 3Department of Health, Medicine and Caring Sciences, Linköping University, 581 85 Linköping, Sweden

**Keywords:** lysosome, acridine orange, lysosomal membrane permeabilization, high throughput

## Abstract

Loss of lysosomal membrane integrity results in leakage of lysosomal hydrolases to the cytosol which might harm cell function and induce cell death. Destabilization of lysosomes often precede apoptotic or necrotic cell death and occur during both physiological and pathological conditions. The weak base acridine orange readily enters cells and accumulates in the acidic environment of lysosomes. Vital staining with acridine orange is a well-proven technique to observe lysosomal destabilization using fluorescence microscopy and flow cytometry. These analyses are, however, time consuming and only adapted for discrete time points, which make them unsuitable for large-scale approaches. Therefore, we have developed a time-saving, high-throughput microplate reader-based method to follow destabilization of the lysosomal membrane in real-time using acridine orange. This protocol can easily be adopted for patient samples since the number of cells per sample is low and the time for analysis is short.

## 1. Introduction

Lysosomes are acidic organelles with an intraluminal pH of 4–5 and over 60 different hydrolases, aiding the degradation of intra- and extracellular molecules [1,2]. In addition to its degradative capacity, the lysosome participates in several other cellular functions including plasma membrane repair via exocytosis, cholesterol homeostasis and cell death regulation [3]. Lysosomal involvement in cell death signaling is well established and occurs via a mechanism called lysosomal membrane permeabilization (LMP), where the release of lysosomal hydrolytic enzymes to the cytosol induces apoptotic or necrotic cell death pathways [4,5,6,7]. In recent years, it has become evident that lysosomal leakage is not always equivalent to induction of cell death [8]. Minor membrane damage induces immediate repair via the ESCRT complexes [9,10], and limited release of cathepsins has been suggested to control different cell functions including cell division, cell movement and inflammation [11]. LMP-induced cell death occurs physiologically as a regulatory mechanism during mammary gland involution [12], but has also been detected in several diseases, such as coronary arteritis [13], fibrotic lung disease [14] and Parkinson’s disease [15]. In addition, we and others have demonstrated that manipulation of lysosomal function could sensitize cancer cells to lysosome-dependent cell death and make them more susceptible to anticancer treatment [16,17,18,19]. Lysosomotropic drugs are weak bases that have lipophilic or amphiphilic properties, allowing them to passively diffuse into the lysosomal lumen [20]. Once diffused, the drug is protonated due to the acidic pH of the lysosome, which impedes the diffusion back to the cytosol. Several drugs used in the clinic have lysosomotropic properties, including antihistamines, antimalarial drugs, antidepressants, antipsychotics, as well as some antibiotics [21]. Some of these drugs have the ability to induce cell death via LMP, which can be utilized as a treatment strategy. For example, the combined use of the cationic amphiphilic antihistamine Loratadine with chemotherapy was shown to reduce mortality in patients with non-small cell lung cancer [22], and the antibiotic azithromycin enhanced the cytotoxicity of DNA-damaging drugs via LMP-induced apoptosis [23]. In addition, lysosomotropic drugs might have a beneficial role in lysosomal storage disorders and neurodegenerative diseases since the increased lysosomal stress activates TFEB and enhances autophagic clearance [24]. Recently, it was suggested that lysosome-targeting drugs could be beneficial for treatment of severe COVID-19 infection [25]. On the other hand, lysosomotropic drugs have also been shown to cause LMP-induced hepatoxicity as a side effect [26,27]. Thus, controlling and analyzing lysosomal stability could have significant importance both clinically and scientifically.

Analysis of lysosomal leakage is often performed as an end-point measurement of lysosomal proteases released to the cytosol, either via immunostaining or immunoblotting of cytosolic fractions [28]. However, these methods are designed to detect damages large enough to allow translocation of proteins and do not detect minor damages, where only smaller molecules such as protons are released. Vital staining using dyes that change their fluorescence properties when the lysosomal pH is altered is a more versatile approach [29]. For this purpose, weak lysosomotropic bases such as acridine orange (AO) can be used. AO (Figure 1A, chemical structure) is a weak base with a pKa of 9.65. In its unprotonated form, AO passively diffuses across membranes and enters the acidic lumen of lysosomes. Once inside, it is protonated by the low pH and is unable to diffuse back, thus accumulating in the lysosome [30]. In comparison to other lysosomotropic dyes, the advantage of AO is its ability to change fluorescence depending on dye concentration. In the cytoplasm, AO exists in a monomeric form that emits green fluorescence. However, when it is concentrated within the lysosome, AO assembles into dimers which creates a metachromatic shift to red fluorescence [31]. In living cells, most of the AO is concentrated inside the lysosomes, which is detected as a punctuate red pattern (Figure 1B, first panel). When the lysosomal proton gradient is lost, leakage of AO to the cytosol can be detected as a concomitant increase in green cytosolic staining and loss of red lysosomal staining (Figure 1B, panel 2–4).

AO is a well-proven staining technique to monitor lysosomal membrane stability, mainly via fluorescence microscopy [32,33] and flow cytometry [34,35]. In two recent studies, we have adapted the AO staining to suit a high-scale approach using a fluorescence microplate reader [16,36]. In this protocol, both the material required and the analysis time is reduced, making it suitable for studying, for example, patient material, as well as making a high-throughput analysis. Also, fluorescence changes are measured over time, which is beneficial compared to the single-point measurement used in flow cytometry.

## 2. Experimental Design

This method has been developed and tested in several different cell types, including human primary melanocytes obtained from fair-skinned donors through foreskin circumcision (0–7 years, parental written informed consent) and four different malignant melanoma cell lines (FM55P, Sigma Aldrich, St Louis, MO, USA; WM115 and WM278, Rockland Immunochemicals, Limmerick, PA, USA; and SkMel28, ATCC, Manassas, VA, USA) [16], murine macrophage-like lymphoma J774 cells (ATCC), patient samples from human alveolar macrophages isolated from bronchoalveolar lavage at Linköping University Hospital as described previously [36], and human foreskin fibroblasts (AG01518, ATCC). The suggested concentrations and conditions presented in this protocol are applicable for all the cell lines tested; however, specific cell culture conditions, medium and concentration of LMP-inducers need to be optimized for each individual cell type.

We have included two known LMP-inducers that can be used as positive controls depending on the experimental system (Figure 2). Leu-Leu methyl ester hydrobromide (LLOMe) is taken up via a dipeptide-specific receptor-mediated endocytosis system [37], although some studies suggest a passive diffusion instead [38]. Once inside the lysosome, LLOMe is converted by cathepsin C to a polymer with membranolytic properties [37,39], acting directly on the lysosomal membrane to cause lysosomal membrane destabilization within minutes after addition. To induce a slower reaction, LMP can be induced indirectly via oxidative stress. In a glucose oxidase (GO)-catalyzed reaction, β-D-glucose is oxidized to gluconic acid using oxygen as a substrate [40], which yields a continuous production of H_2_O_2_. After entering the lysosome, H_2_O_2_ is reduced by lysosomal iron in the Fenton reaction, and the produced hydroxyl radical initiates lipid peroxidation [41,42]. The lipid peroxidation in turn destabilizes the lysosomal membrane and induces LMP.

### 2.1. Considerations When Designing the Experiment

#### 2.1.1. Lysosomal pH, Volume and Numbers

Similar to other lysosomal dyes, the limitation of the AO method lies in its pH-dependent accumulation within acidic vacuoles. AO only accumulates inside the lysosome when the pH is low, and the accumulation can thus be affected by agents that cause a shift in pH. In addition, there are several treatments known to cause changes in lysosomal volume, numbers, etc., affecting the intensity of initial AO staining and thus impeding the interpretation of the results. For example, compounds that inhibits autophagy or alter lysosomal function can cause an upregulation of lysosomal biogenesis through activation of TFEB, or increased endosomal–lysosomal fusion, in order to compensate for the loss of function [24]. Therefore, it is important to microscopically evaluate if any pretreatment applied before LMP induction affects lysosomal characteristics and/or AO staining, and to be cautious if initial fluorescence values in the plate reader is affected. In the data analysis section below, we use normalization to the initial fluorescence value for each sample to eliminate variation in staining intensities. This should, however, always be carried out with caution when treatments affect lysosomal characteristics.

#### 2.1.2. Phototoxicity and Photobleaching

Several studies have shown that AO has phototoxic properties when accumulated inside the lysosome, which also can be used to induce LMP (Figure 1B). If cells are subjected to prolonged illumination with blue light, AO sensitizes the lysosomal membrane to photo-oxidation, resulting in loss of the proton gradient and release of the dye, generating a shift to the green fluorescence spectra [43,44]. Thus, high microscopy skills are needed to avoid this phenomenon and to keep the time of finding focus as short as possible during live cell microscopy of AO-stained cells. In the plate reader system, the cells are only exposed to the blue light for a very short time compared to the microscope, and we do not detect any shift from red to green fluorescence in the control cells. Therefore, the plate reader method appears to be less phototoxic. However, we do detect a slight decrease in green fluorescence, which could be attributed to photobleaching over time [45] or diffusion of monomeric AO out of the cell.

#### 2.1.3. AO Staining of RNA and DNA

It is important to note that AO also can intercalate with RNA and DNA strands. This is the case in fixed cells, where the non-ionized AO binds densely to the acidic parts of single-stranded RNA, and more loosely to double-stranded DNA, thus staining RNA red and DNA green. However, in living cells, AO does not intercalate at all with DNA due to chromatin inaccessibility, and only sparsely with cytoplasmic and nucleolar RNA, generating a weak and diffuse green signal [46,47]. Instead, it readily diffuses into the acidic lysosome where it becomes ionized and trapped, and thus fluoresces in red. In cells where lysosomal membrane integrity is lost, the loss of proton gradient allows non-ionized AO to passively diffuse across membranes and bind to cytoplasmic and nucleolar RNA [48], attributing to the increased green signal seen after LMP induction.

### 2.2. Materials

Adherent primary cells or cell line of choice.Dulbecco’s Modified Eagle’s Medium (DMEM) supplemented with 10% fetal bovine serum (FBS), 100 IU/mL penicillin and 100 µg/mL streptomycin (all from Gibco, Paisley, UK), or other complete cell culture medium suitable for the cells of choice.Phenol-free DMEM supplemented with 10% FBS, 100 IU/mL penicillin and 100 µg/mL streptomycin, or other phenol-free complete cell culture medium.Acridine orange (Sigma-Aldrich, Cat. No.: 318337).Positive control 1: LLOMe (Sigma-Aldrich, Cat. No.: L7393).Positive control 2: GO (Sigma-Aldrich, Cat. No.: G2133).96-Well Black Polystyrene Microplate with clear bottom (Corning, Cat. No.: 3603).

### 2.3. Equipment

Fluorescence plate reader with heating, equipped with excitation/emission filter at 485/535 nm for green fluorescence and 465/650–710 nm (see Section 4 for choosing the correct emission wavelength) for red fluorescence. We kept the measurement time as short as possible. We have used a Spark 10 M reader (Tecan, Männedorf, Switzerland) with an integration time of 40 µs.

## 3. Procedure

### 3.1. Preparation of Stock Solutions

Prepare stock solutions of the following:1 mg/mL AO in H_2_O. Store at 4 °C for up to 6 months in the dark;0.1 M LLOMe in H_2_O. Store at −20 °C for up to 6 months;1 mg/mL GO in 50 mM sodium acetate. Prepare fresh solution for each experiment.

### 3.2. Cell Seeding

Grow cells in DMEM supplemented with 10% FBS, 100 IU/mL penicillin and 100 µg/mL streptomycin, and culture under normal conditions at 37 °C in 5% CO_2_.

For experiment, use trypsinize cells and seed in a black 96-well plate for fluorescence reading. Use a cell density to allow 80–90% confluence at the day of experiment.Make triplicates of each sample and include at least one well without cells for measurement of background fluorescence.Allow cells to adhere for at least 24 h.


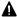
 Proper optimization of cell density vs. drug concentration must be performed before the start of the experiment. Also, low density of cells can generate uncertain intensity values, especially for red fluorescence.

### 3.3. Staining of Cells

At start of experiment, stain cells with AO following the steps mentioned below:Prepare AO staining solution by diluting the AO stock in complete cell culture medium to reach a final concentration of 2–5 µg/mL;Remove culture medium from the cells and add 100 µL staining solution/well;Incubate for 15 min at 37 °C;Wash cells with 100 µL complete phenol-free medium for 2 × 5 min;Leave cells in 100 µL complete phenol-free medium. Remember to add medium to the well without cells.

### 3.4. Measurement of Initial AO Fluorescence

Preheat the instrument to 37 °C.Set to measure green fluorescence at excitation 485 nm and emission 535 nm. If red fluorescence is to be analyzed, use excitation 465 nm and emission 650–710 nm (see Section 4 for choosing the optimal emission wavelength).Be aware that green fluorescence is markedly increased upon LMP, thus, be cautious for signal saturation when setting the gain.Make a single-point measurement to record initial AO fluorescence before induction of LMP.

### 3.5. Induction of Lysosomal Leakage

Induce LMP using the compound of interest. Optional: include one of the following as a positive control (the appropriate concentration of the LMP inducers must be determined for the selected cell type. The concentration spans suggested below corresponds to the cell types in our studies):5–50 µg/mL glucose oxidase;0.1–5 mM LLOMe.

### 3.6. Measurement of AO Fluorescence

Start the collection of data for fluorescence change over time at 37 °C immediately after the addition of LMP inducer. Use the same settings as for initial AO fluorescence measurement.The time between each measurement can vary depending on the LMP inducer. For lysosomal membrane-targeting drugs, such as LLOMe, maximal leakage usually occurs within 30 min and fluorescence intensity data are collected every minute. Indirect LMP inducers such as GO may need up to 2 h of data collection and data can be collected every 2nd–3rd min.

### 3.7. Data Analysis

Subtract the background.Divide the fluorescence value from each time point with the initial fluorescence value to obtain a ratio of fluorescence change.Calculate the mean value of the triplicates for each time point.Plot fluorescence intensity change over time (green curve in Figure 3).In order to compare LMP of several treatments, calculations of relative fluorescence unit per second (RFU/s) might be useful: oUse the plot of relative fluorescence over time as obtained above and define the highest slope of the curve (dotted blue line in Figure 3);oCalculate $\frac{∆fluorescence}{∆time}$ by subtracting the fluorescence value for the first time point (F1) to the fluorescence value for the end time point (F2) for the defined time interval. Divide the obtained value with the time interval (t2–t1) to obtain RFU/s;oUse the same time interval for all samples within one experiment.
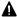
 We have observed a slight reduction in fluorescence over time in control cells not exposed to LMP inducers. Therefore, we recommend to always include unexposed controls for all conditions used in the experiment.


## 4. Expected Results

In Figure 4A, we show the emission spectra for AO in control and GO-exposed J774 cells, measured in a SPARK 10 M plate reader. Control cells have two emission peaks, one at 535 nm representing the monomeric AO which fluoresces in green, and one at 650 nm representing the dimeric red AO inside the lysosome. In cells exposed to GO for 90 min, the green peak at 535 nm is much higher as a result of leakage of AO out from the lysosome. The peak at 650 nm is not visible in these cells due to the loss of lysosomal dimeric AO, and it is apparent that the red fluorescence is lower compared to control cells. In the literature, AO red fluorescence is often measured at 650 nm, but we have discovered that high concentrations of LMP inducers can result in an incorrect reading after LMP induction, where the red fluorescence does not decrease or even increases slightly. Increase in the red channel is highly unlikely and probably the result is due to bleed-through from the green channel, since it greatly increases over time. In this occasion, we find a more prominent decrease in red fluorescence when measuring AO at higher wavelengths. Since the difference in fluorescence intensity between the control cells and GO-exposed cells remain until ~720 nm, it is possible to analyze them at a higher wavelength and we recommend this approach. However, as shown in Figure 4B,C, the decrease in red intensity is not as readily detected as the increase in green fluorescence. When a lysosomotropic dye display high fluorescence intensity in the lysosome, the sensitivity to monitor a minor decrease in fluorescence is reduced. Therefore, increase in AO green fluorescence intensity offers higher sensitivity and will better reflect alterations in a lysosomal proton gradient.

By analyzing the change in AO green fluorescence over time, concentration-dependent lysosomal leakage can be detected (Figure 5A). In human malignant melanoma WM278 cells stained with AO and exposed to increasing concentrations of LLOMe, lysosomal leakage occurs almost immediately, and the magnitude varies depending on LLOMe concentration. To validate the results, we also stained cells using LysoTracker, another lysosomotropic dye that is widely used to detect lysosomes [45,49]. LysoTracker is dependent on pH, and only stain intact acidic lysosomes. Live cell imaging, before and 1 min after LLOMe exposure, confirm that lysosomal membrane damage occurs immediately in cells exposed to higher concentrations of LLOMe (Figure 5B).

Time study of AO-stained human fibroblast AG01518 cells exposed to LLOMe show that lysosomal membrane integrity is almost completely abolished within 5 min (Figure 5C), which is confirmed in LysoTracker-stained cells (Figure 5D). Both AO and LysoTracker are sensitive to small perforations in the lysosomal membrane that allows the release of protons, a feature that often precedes the release of larger molecules [9,10]. To illustrate the alternative methods for LMP detection and underline the rapid response of AO, we present the results from galectin-3-mediated lysosomal repair [50] and release of lysosomal cathepsin to the cytosol. Detection of damaged lysosomes using punctuate galectin staining has emerged as a popular method to identify lysosomes with larger perturbations. Galectins are recruited to damaged lysosomes, where they bind to glycosylated lysosomal membrane proteins that are exposed upon lysosomal damage [51]. As shown in Figure 5E, galectin-3 puncta, although evident to a minor extent after 15 min, first increases substantially after 1 h of LLOMe exposure, as does the cytosolic release of mature cathepsin D (Figure 5F). The release of cathepsin D is concentration-dependent, where 0.1 mM only causes a minor release, while 0.5 mM and 1 mM is more substantial after 1 h of LLOMe (Figure 5G). This demonstrates that pH-sensitive markers such as AO and LysoTracker are more sensitive to early occurring, small membrane perforations, while galectin-3 marking and cathepsin D release detect larger lysosomal damage, which often occur at later timepoints.

## 5. Conclusions

The toolbox for studying and estimating lysosomal stability is constantly increasing, which also mirrors the expanding interest for lysosomal function. Several methods to quantify and visualize LMP have been presented over the years and a careful choice of analysis method will provide information about the extent and speed of damage. Release of lysosomal contents to the cytosol has been estimated through the analysis of lysosomal enzyme activity in the cytosol [52,53] and monitored via immunoblotting [54]. LMP has been characterized with the presence of lysosomal proteases in the cytosol, detected using immunofluorescence or immunoelectron microscopy, as well as by the release of fluorophore conjugated dextran to the cytosol [7,28]. Recently, it was discovered that limited lysosomal damage, only allowing the release of small molecules, can be repaired via an ESCRT-mediated mechanism [9,10]. For larger damage, repair is not plausible, instead the cell activates the lysophagic machinery to clear the entire damaged organelle [9,55]. This emphasizes the need for several assays to follow time-dependence as well as repair during LMP-induced damage, and also to detect damage that may not always result in the release of cathepsins and/or cell death. The presented microplate analysis for high-throughput AO staining can be used to detect minor insults to the lysosomal damage and for continuous measurement of lysosomal stability. It is also a method that is suitable for, e.g., screening of drug libraries and patient samples, due to the high speed and low requirement of material.

## Figures and Tables

**Figure 1 mps-06-00072-f001:**
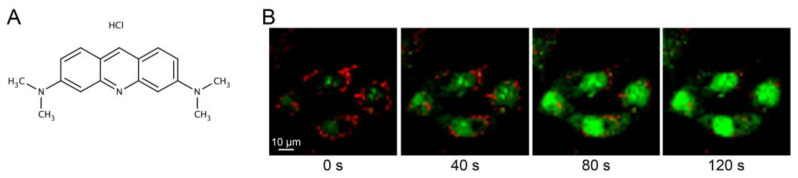
(**A**) Chemical structure of acridine orange. (**B**) Live cell imaging using a Zeiss LSM 700 confocal microscope (40× objective) with the Zen software of acridine orange-stained fibroblasts exposed to photo-oxidation via blue light (488 nm) to induce LMP. Fibroblasts show accumulation of AO in intact lysosomes (0 s), and photo-oxidized induced LMP causes AO leakage and emission of green fluorescence (40–120 s).

**Figure 2 mps-06-00072-f002:**
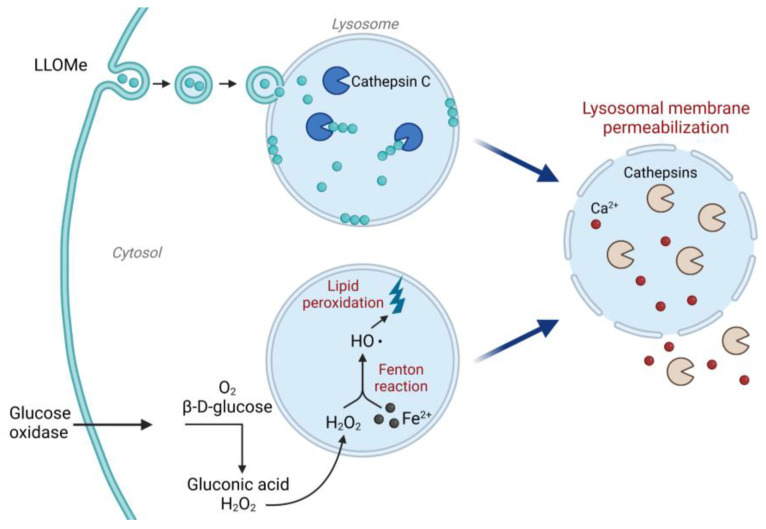
Mechanism of action for LLOMe and glucose oxidase. LLOMe is taken up via endocytosis and converted by cathepsin C in the lysosome to a membranolytic polymer. The LLOMe polymer acts directly on the lysosomal membrane to trigger lysosomal membrane permeabilization and release of lysosomal content. Glucose oxidase produces H_2_O_2_ by catalyzing the oxidation of β-D-glucose to gluconic acid. H_2_O_2_ enters the lysosome via diffusion and can then react with lysosomal iron through the Fenton reaction, inducing oxidative stress and lipid peroxidation, which cause lysosomal membrane damage. Image created with BioRender.com (accessed on 1 August 2023).

**Figure 3 mps-06-00072-f003:**
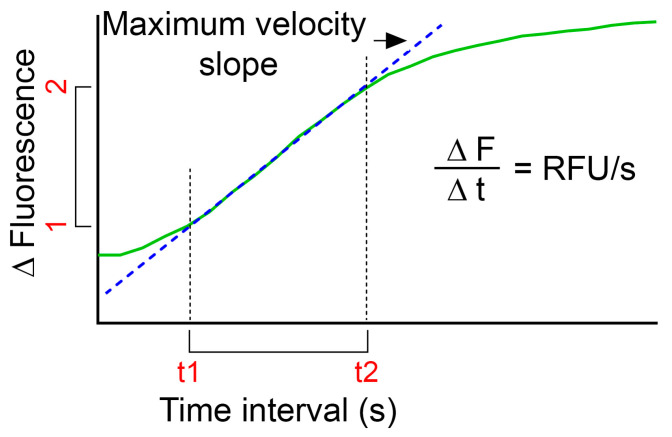
Graph depicting calculation of RFU/s. Green line is the fluorescence intensity change over time and the dotted blue line represents the maximum slope of the curve.

**Figure 4 mps-06-00072-f004:**
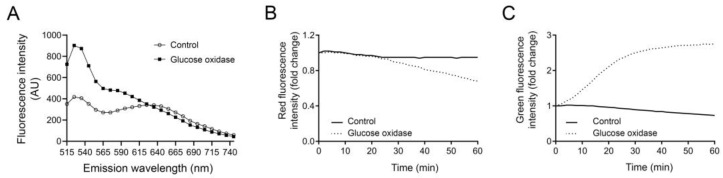
(**A**) Emission spectra of acridine orange (Ex 465 nm) in J774 controls and cells treated with glucose oxidase for 90 min. Changes in acridine orange, (**B**) red fluorescence (Em 710 nm) and (**C**) green fluorescence (Em 535 nm) in cells exposed to 18 µg/mL glucose oxidase (0–60 min).

**Figure 5 mps-06-00072-f005:**
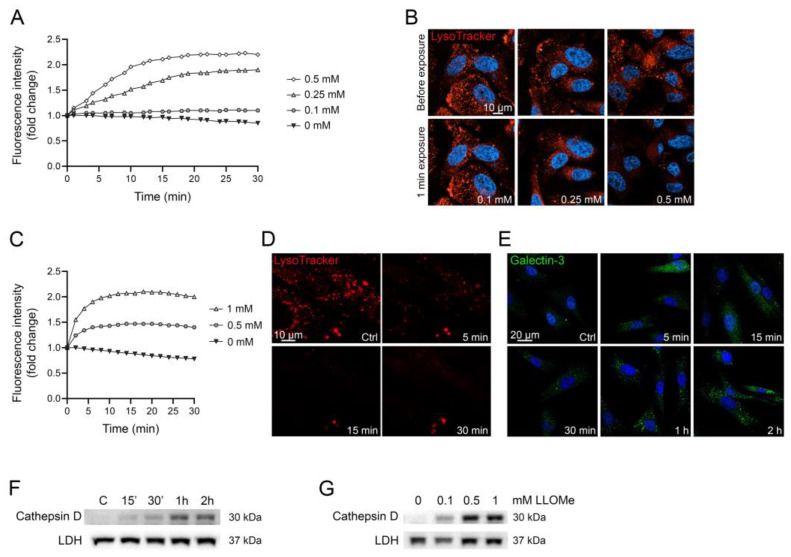
WM278 malignant melanoma cells (**A**,**B**) and AG01518 human fibroblasts (**C**–**G**) were exposed to 0–1 mM LLOMe. (**A**) Changes in acridine orange green fluorescence over time. (**B**) Live cell images before and after LLOMe exposure (1 min) of LysoTracker-stained cells (100 nM, 30 min). (**C**) Changes in acridine orange green fluorescence over time. (**D**) Live cell imaging of LysoTracker-stained cells (100 nM, 30 min) before and after LLOMe exposure (1 mM). (**E**) Immunostaining of galectin-3 (BD Pharmingen #556904, San Diego, CA, USA, 1:100) in cells exposed to 1 mM LLOMe. (**F**) Immunoblotting of cathepsin D in cytosolic fractions showing time-dependent increase in cells exposed to 1 mM LLOMe and (**G**) concentration-dependent increase after exposure to LLOMe for 1 h. Unedited immunoblots in Appendix A. Images were obtained using a Zeiss LSM 700 (**D**,**E**) or 800 (**B**) confocal microscope (40× objective, Zen Blue 3.1 software).

## Data Availability

The data that support the findings of this study are available from the corresponding author, (IE), upon reasonable request.

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
