# Peer review of "Real-Time Monitoring of Lysosomal Membrane Permeabilization Using Acridine Orange"

_mps, 2023, doi:10.3390/mps6040072_

Round 1

Reviewer 1 Report

The main aim of the presented manuscript entitled "Real-time monitoring of lysosomal membrane permeabilization using acridine orange" was a presentation of the method developed by the authors to assess the destabilization of the lysosomal membrane, which is based on the microplate technique and the determination of the fluorescence intensity of acridine orange (a lysosomotropic compound).

Below are comments on the peer-reviewed manuscript.

 General comments:

1. Introduction. I propose to supplement/extend this part of the manuscript with the importance of lysosomal membrane permeability in various therapies where lysosomotropic drugs are used. This will increase the importance and purposefulness of the developed method.

2. Experimental Design. I suggest supplementing this part with a diagram showing the mechanism of lysosomal membrane permeability by the LMP inducers proposed in the methodology. The source/origin of the cells used in the study and the names of the cell lines should be provided.

3. 2.1.1. The authors wrote in too general terms about the factors affecting the intensity of the initial AO staining, and thus hindering the interpretation of the results. They should be replaced.

"In addition, there are several treatments known to cause changes in lysosomal volume, numbers etc., affecting the intensity of initial AO staining and thus impeding the interpretation of the results."

4. 2.1.2. Please explain the statement that greater phototoxicity is observed when using microscopy to image lysosomes in cells than when using a plate reader.

5. The manuscript lacks literature, therefore it is difficult to fully refer to the entire manuscript, because it is not known if literature items have been correct cited.

Other comments:

1. Keywords. I propose to write: lysosomal membrane permeabilization, instead of membrane permeabilization.

2. In the methodology (2.2. Materials), please complete the origin (company) of the culture media and antibiotics.

3. In Figure 1 there is no scale in the photos or information in the description of the photos about the magnification at which they were taken. There is also no information about the type of microscope (camera/software) used in the study (Figures 1 and 4).

4. Please explain why in the methodology includes the range of concentrations of the LMP inducers used, i.e. 5 - 50 µg/ml glucose oxidase, 0.1 - 5 mM LLOMe. For example, in figure 3 the concentration of GO used is 18 µg/ml.

5. There are no concentrations/solvents for the dyes used: LysoTracker and galectin-3.

6. In Figure 3 (graph A) the mark on the X-axis is missing.

Reviewer 2 Report

In their manuscript, Eriksson and coworkers describe the adaptation of the well-established acridine orange (AO) test for the detection of lysosomal membrane permeabilization (LMP) to a small scale, high throughput system. This new version of the method can be performed on a 96-well plate, thus reducing the number of cells for the assay and the volume of reagents, and allows the simultaneous processing and analysis of a great number of samples. Moreover, such a version of the AO technique coupled to the use of an automated plate reader is suitable for real-time LMP detection.

The manuscript is basically well written, the experimental section has sufficient details to allow the use of the protocol and the adaptation to the cellular systems and the instruments available in the different laboratories. However, I have found and listed below some aspects that would need further attention by the Authors.

Major concerns

1. The AO technique in its different versions, namely the uptake and relocation methods, is in use since many years and its validity has been demonstrated in several systems and experimental conditions. However, the molecular mass of AO is very small compared with that of the lysosomal proteases that, once released to the cytoplasm, may trigger lysosomal cell death. Thus, on one hand, the smaller the tracker is, the greater is the sensitivity of the technique to reveal an even limited LMP. On the other hand, whether this ‘minimal’ LMP is sufficient to produce any biological effect or not is far from being demonstrated. In several cases, in fact LMP occurs at very low levels and is only a transient event, which precedes lysosomal membrane repair; on its own, it thus may or may not lead to the release of the lysosomal contents and lysosomal cell death. As the Authors certainly know, this fact has been clearly proved with the use of LLOMe, the lysosomotropic agent used to set up the technique. In my opinion, these aspects should be clearly recalled and presented as one of the potential limitations to the use of the AO protocol to reveal LMP.

2. Lines 77-78. The text says that ‘… Leu-Leu methyl ester hydrobromide (LLOMe) acts directly on the lysosomal membrane by entering the lysosome through receptor-mediated endocytosis…’. This is a misinterpretation of the original cited research from Uchimoto et al., in which LLOMe was shown to enter the plasma membrane, not the lysosomal one, by a receptor-mediated endocytosis. It should be noted that a later research by Repnik et al. (2017; doi:10.1242/jcs.204529) states that LLOMe ‘…passively crosses cell membranes into the cytoplasm and accumulates in lysosomes either because of protonation in the acidic lumen or alternatively because of ester hydrolysis…’. Although not the focus of the manuscript, the text must be revised to amend the errors and to provide the readers with the most accurate information.

3. Lines 222-223. …‘AO red fluorescence is often measured at 650 nm, but we have discovered that this occasionally results in an incorrect increased reading after LMP induction…’. The authors should clarify this point. What treatment to induced LMP was actually increasing Rred AO fluorescence? In what conditions? Can the authors exclude that the treatment led to osmotic swelling of lysosomes, resulting thus in an increased AO accumulation and red fluorescence?

4. Line 240. The Authors state that the ‘…lysosomal leakage occurs almost immediately, and the magnitude varies depending on LLOMe concentration.’ Fig. 4A shows that LMP due to 0.5 and 1 mM LLOMe perfectly overlap. Does this fact mean that the lysosomes are already totally permeabilized by 0.5 mM LLOMe or that the technique is effective only up a given level of LMP? In the same way, is there any reason why Lysotracker Red fluorescence is almost completely lost with 0.5 mM LLOMe, but is still appreciable with 1mM, whilst one could expect the opposite?

Minor concerns

1.    Line 166: is phenol-free medium essential? To the best of my knowledge, phenol red-containing medium neither affect AO uptake nor impairs fluorescence emission/detection. Can the Authors comment?

Round 2

Reviewer 2 Report

I have no further comments or requests.